# Multiple Organic Contaminants Determination Including Multiclass of Pesticides, Polychlorinated Biphenyls, and Brominated Flame Retardants in Portuguese Kiwano Fruits by Gas Chromatography

**DOI:** 10.3390/foods12050993

**Published:** 2023-02-26

**Authors:** Virgínia Cruz Fernandes, Martyna Podlasiak, Elsa F. Vieira, Francisca Rodrigues, Clara Grosso, Manuela M. Moreira, Cristina Delerue-Matos

**Affiliations:** REQUIMTE/LAQV, Instituto Superior de Engenharia do Porto, Instituto Politécnico do Porto, Rua Dr. António Bernardino de Almeida 431, 4249-015 Porto, Portugal

**Keywords:** pesticides, environmental contaminants, QuEChERS, gas chromatography, kiwano

## Abstract

Global production of exotic fruits has been growing steadily over the past decade and expanded beyond the originating countries. The consumption of exotic and new fruits, such as kiwano, has increased due to their beneficial properties for human health. However, these fruits are scarcely studied in terms of chemical safety. As there are no studies on the presence of multiple contaminants in kiwano, an optimized analytical method based on the QuEChERS for the evaluation of 30 multiple contaminants (18 pesticides, 5 polychlorinated biphenyls (PCB), 7 brominated flame retardants) was developed and validated. Under the optimal conditions, satisfactory extraction efficiency was obtained with recoveries ranging from 90% to 122%, excellent sensitivity, with a quantification limit in the range of 0.6 to 7.4 µg kg^−1^, and good linearity ranging from 0.991 to 0.999. The relative standard deviation for precision studies was less than 15%. The assessment of the matrix effects showed enhancement for all the target compounds. The developed method was validated by analyzing samples collected from Douro Region. PCB 101 was found in trace concentration (5.1 µg kg^−1^). The study highlights the relevance of including other organic contaminants in monitoring studies in food samples in addition to pesticides.

## 1. Introduction

The consumers’ interest in new and exotic fruits has intensified, mainly due to the growing knowledge regarding their bioactive composition and biological activities with pro-healthy effects. Kiwano (*Cucumis metuliferus* E. Mey), belonging to the Cucurbitaceae family, is a plant naturally occurring in South Africa, Nigeria, Namibia, Botswana, and Southern Sahara, being also sporadically found in Yemen [1]. In the last years, its exportation has grown in countries such as Kenya, New Zealand, France, and Portugal [1,2].

The ripe kiwano fruit is characterized by an orange skin with many blunt thorns on its surface and green, jelly flesh inside [1,2,3,4]. Kiwano fruit has low levels of carbohydrates and calories but high contents of water, minerals including magnesium, calcium, potassium, iron, phosphorus, zinc, copper, and complex B vitamins, vitamin C, and β-carotene [1,2]. Some pharmacological properties of this exotic fruit have been recently revised by Vieira et al. [3], including anticardiovascular, antidiabetic, antiulcer, antioxidant, anti-inflammatory, antimalarial, and antiviral activities. 

Due to these beneficial properties, its production, exportation, and consumption have increased, leading to intensive cultivation. As such, these particular fruits contribute directly and importantly to food security and nutrition in most producing zones, however, some food safety issues are still little explored in these matrices. There are several ways of improving plant cultivation. One of them is the use of plant protection products, commonly known as pesticides, which may have a chemical source as well as a natural origin [5]. Pesticides are used to protect crops from the harmful activity of other plants, microorganisms, insects, or even animals [6]. Although higher yields of cultivation can be obtained by using pesticides [7], they represent a threat to animals and human health and lives. Other toxic chemical substances that are present in the environment due to man-made activity derived from different sources (e.g., plastics, industrial, etc.), are referred to as environmental pollutants (e.g., polychlorinated biphenyls (PCB), polybrominated diphenyl ethers (PBDE), polycyclic aromatic hydrocarbons (PAH), heavy metals). Many of these compounds can be resistant to environmental degradation and accumulate in soil and food [8]. 

Further, prolonged exposure to these agricultural chemicals, particularly by contaminated food consumption, may lead to chronic disorders, such as cancer, hormone disruption, diabetes, asthma, or infertility [9,10,11] and neurodegenerative disorders [12]. As an example of the pesticide family, organophosphorus pesticides (OPP) are highly toxic chemical compounds used as insecticides for crop protection [13]. These chemicals are neurotoxic, as they inhibit acetylcholinesterase (AChE), which causes malfunctions in muscular activity leading to seizures, paralysis, or even death [14]. Further, persistent organic pollutants (POP), including organochlorine pesticides (OCP), PCB, PBDE, and PAH, are organic lipophilic chemicals that bioaccumulate in fatty tissues, also causing adverse effects on human health and the environment [15,16]. Exposure to POP is associated with malfunctions in the reproductive and endocrine systems [17], being also responsible for the development of many cancer types. Apart from human health, the use of pesticides is deleterious to the environment. Because of this, many flora and fauna species are exposed to multiple contaminants. Water, soil, and air pollution caused by the use of chemicals leads to disturbances in the ecosystem and poses a threat to biodiversity [5,18]. Therefore, their use must be restricted [19]. Due to the toxicity of environmental pollutants, their content needs to be continuously monitored, and attention to them is crucial. Besides that, surveys of pesticide residues in fruit are important to validate conformity with strict regulations of newly open markets for the exportation of exotic fruit. 

The European Commission establishes the maximum residue levels (MRLs) for pesticides to minimize the exposure of humans to harmful levels in food or feed [20]. Pesticides and several environmental pollutants have been reported in the literature on food [21,22,23,24,25,26,27,28]. However, there is a lack of studies regarding new fruits that are not yet legislated even though there is a high demand, and environmental contaminants are also not legislated [29,30,31]. Even more, one of the ambitious goals set by the European Green Deal and the Farm to Fork Strategy includes a 50% reduction in the use of pesticides by 2030. 

This strikes a challenge to analytical chemistry, namely in the development and validation of sensitive analytical methods. One of the best approaches for multiresidue analysis (simultaneously pesticide and other contaminants) in food samples is the extraction by Quick, Easy, Cheap, Effective, Rugged, and Safe (QuEChERS) method [32]. It is a very convenient, time- and reagent-saving solid-phase extraction-based procedure consisting of two major steps [33]. In the first step, the fruit, vegetable, or other food sample is subjected to extraction with acetonitrile (MeCN) and salts (e.g., MgSO_4_, NaCl), followed by a second step in which a sample clean-up via dispersive solid-phase extraction (d-SPE) is performed [20]. Afterward, the extracted and purified compounds are commonly analyzed with the use of gas chromatography (GC)-based methods [34]. Particularly, GC coupled with a mass spectrometer (MS) is favored for such a complex multiple contaminants identification due to the low limits of detection (LOD) [35]. Tandem mass spectrometry, specifically GC-MS/MS and LC-MS/MS, and other selective detectors were reported to be more efficient in simultaneously detecting multiple contaminants [36].

Considering the beneficial properties associated with the kiwano and its increasing consumption, it becomes urgent to develop methodologies and evaluate this fruit’s safety [37]. To the best of our knowledge, there are no analytical methods developed or monitoring studies that report the chemical safety in terms of pesticides and other environmental contaminants, namely plastic-related chemicals and others associated with anthropogenic sources, in kiwano fruit samples. Therefore, the aim of this study was to optimize and validate an extraction methodology for the simultaneous analysis of 30 multiple contaminants (6 OPP, 12 OCP, 5 PCB, and 7 BFR) from kiwano fruit samples using QuEChERS method and d-SPE clean-up to detect trace levels of these contaminants using GC techniques.

## 2. Materials and Methods

### 2.1. Reagents and Standards

Analytical standards of high purity (≥97%) for seven brominated flame retardant (BFR) compounds (2,4,4′-tribromodiphenyl ether (BDE28), 2,2′,4,4′-tetrabromodiphenyl ether (BDE47), 2,2′,4,4′,5-pentabromodiphenyl ether (BDE99), 2,2′,4,4′,6-pentabromodiphenyl ether (BDE100), 2,2′,4,4′,5,5′-hexabromodiphenyl ether (BDE153), 2,2′,4,4′,5,6′-hexabromobiphenyl ether (BDE154), and 2,2′,4,4′,5,5′-hexabromodiphenyl ether (BDE183)) were obtained from Isostandards Material, S.L. (Madrid, Spain). The five PCB standards (2,4,4′-trichlorobiphenyl (PCB28), 2,2′,4,5,5′-pentachlorobiphenyl (PCB101), 2,3′,4,4′,5-pentachlorobiphenyl (PCB118), 2,2′,4,4′,5,5′hexachlorobiphenyl (PCB153), and 2,2′,3,4,4′,5,5′-heptachlorobiphenyl (PCB180)) were acquired from Riedel-de Haën (Seelze, Germany). The eighteen pesticides with analytical grade (12 OCP (hexachlorobenzene (HCB), α-, β-, and ζ-hexachlorocyclohexane (HCH), [1,1,1-trichloro-2-(2-chlorophenyl)-2-(4-chlorophenyl)ethane] (o,p′-DDT), 2,2-bis(4chlorophenyl)-1,1-dichloroethylene (p,p′-DDE), 1-chloro-4-[2,2dichloro-1-(4-chlorophenyl)ethyl]benzene (p,p′-DDD), aldrin, dieldrin, α-endosulfan, methoxychlor, and lindane) and 6 OPP (chlorfenvinphos, chlorpyrifos, chlorpyrifos-methyl, dimethoate, parathion-methyl, and malathion) were obtained from Sigma-Aldrich (St. Louis, MO, USA). The internal standards (IS) 4,4′-dichlorobenzophone and triphenyl phosphate were from Sigma-Aldrich (St. Louis, MO, USA). QuEChERS extraction kits, clean-ups, and SampliQ GCB (Graphitized carbon black) SPE Bulk Sorbent were from Agilent Technologies (Santa Clara, CA, USA). Chromatography grade *n*-hexane and acetonitrile (MeCN) were purchased from Merck (Darmstadt, Germany) and Carlo Erba (Val de Reuli, France), respectively. Ultrapure water (UPW) with water sensitivity >18.2 MΩ⋅cm at 25 °C was produced with a Milli-Q water purification system (Millipore, MA, USA).

### 2.2. Samples

Ten kiwano fruits were supplied by a local farm located at Cinfães, Douro, Portugal. The mature fruits were collected in February 2019 from 10 different plants (random sampling) to obtain a representative set of fruits. The pulp of kiwano was separated from the orange skin, ground in a miller, homogenized, and finally, stored at −18 °C. 

### 2.3. Extraction Procedure: Optimization and Validation

The 30 multiple contaminants were extracted from the kiwano samples based on the previously reported QuEChERS method with d-SPE clean-up [22]. The procedure, whose schematic illustration is shown in Figure 1, included five steps: (1) 5 g of kiwano pulp sample was weighed into a 50 mL polypropylene tube, (2) 8 mL of MeCN and 2 mL of UPW were added, and the tube was thoroughly vortexed for 1 min, EN QuEChERS (4 g MgSO_4_, 1 g NaCl, 1 g NaCitrate, 0.5 g disodium citrate sesquihydrate) were added, the tubes were shaken for 1 min with a vortex, and centrifuged for 5 min at 2490 rcf at room temperature, (3) 1 mL of the supernatant was transferred to the 2 mL d-SPE clean-up tube (150 mg of MgSO_4_, 50 mg of PSA, and 25 mg of GCB) and the tubes were vortexed for 1 min and centrifuged for 5 min at 2490 rcf at room temperature, (4) 900 µL of the final extract was transferred to a labelled vial, the extract was dried under nitrogen flow, and it was redissolved in 900 µL of *n*-hexane, and finally, (5) the sample was vortexed and 150 µL of the extract with the addition of 100 µg L^−1^ of the IS was added in the vial and was placed in the autosampler for the gas chromatography (GC) analysis. The IS was used to control the analytical quality of the GC analysis. Extractions were performed in triplicate.

For the optimization of the methodology, pre-spiking and post-spiking experiments were carried out to evaluate the extraction efficiency. The procedure for pre-spiking was the same as described above (Figure 1), with the difference that the sample in step 1 was contaminated with 7.5 µg kg^−1^ from the mixture of 30 multiple contaminants. The following steps remained the same, as shown in Figure 1. The procedure for the post-spiking had a change in step 4. Before injection in the GC, 7.5 µg kg^−1^ of the 30 multiple contaminants was added to the vial and redissolved in the kiwano fruit extract. The extraction efficiency was studied in terms of recoveries percentages comparing the results obtained between the pre-spiking and post-spiking studies. 

The validation of the method developed was performed following the Eurachem guidelines and SANTE/11312/2021 document by studying several analytical parameters, such as the linearity, recovery at three spiking levels (7.5, 11.2, 14.9 µg kg^−1^) and 5 replicates matrix effects, and intra-day and inter-day precision (experiments with the 7.5 µg kg^−1^ spiking level by five repeated measurements in the same and intercalary days). Quantification was performed using matrix-matched calibration (linearity between 1.5–18.7 µg kg^−1^) and solvent calibration (linearity between 10–125 µg L^−1^). The analytical validation was performed in the GC coupled to an electron capture detector (GC-ECD) and GC coupled to a flame photometric detector (GC-FPD), and with the regression analysis, the linearity was evaluated, and the limits of detection and quantification (LOD and LOQ) were determined. 

### 2.4. Equipment

The GC analysis was performed according to Dorosh et al. [22]. Briefly, the halogenated organic compounds (5 PCB, 7 BFR, and 12 OCP) were analysed using GC-ECD (GC-2010, Shimadzu, Quioto, Japan) and OPP using a GC -FPD (GC-2010, Shimadzu, Quioto, Japan). The presence of contaminants was confirmed by GC/MS. Confirmation was based on a comparison of sample GC retention time and product ion abundance ratios (mass to charge ratio, *m*/*z*) against those obtained for a reference standard. The system control and the data acquisition were performed in Shimadzu’s GC Solution software in GC-ECD and GC-FPD and Xcalibur software in GC/MS. The GC analysis was performed in triplicate. 

#### 2.4.1. GC-ECD

The analysis was performed using a capillary GC column Zebron-5MS (30 m × 0.25 mm × 0.25 μm) (Phenomenex, Madrid, Spain). The oven temperature was programmed at 40 °C for 1 min, increased to 120 °C at a rate of 15 °C/min where it was kept for 1 min. Then, the temperature was increased once more at a rate of 10 °C/min to 200 °C, where it was kept for 1 min, and lastly, the temperature was increased from 7 °C/ min to 290 °C and held for 10 min. The injection was performed in splitless mode. The temperatures of the injector and ECD were 250 °C and 300 °C, respectively. Helium was used as a carrier gas (1.3 mL/min), and nitrogen as a makeup gas (30 mL/min). 

#### 2.4.2. GC-FPD

The GC-FPD column was the same as the one described in Section 2.4.1. The carrier gas was helium at 1 mL/min with a linear velocity of 25.4 cm s^−1^. The detector was at 250 °C in injection was performed in splitless mode, and the analytes were detected at 290 °C. The column was programmed at 100 °C, which was kept for 1 min before increasing it to 150 °C at a rate of 20 °C/min, where it was held for 1 min. Following, the temperature was increased to 180 °C at 2 °C/min and kept for 2 min, and finally, increased at 20 °C/min to 270 °C, where it was kept for 1 min. 

#### 2.4.3. GC/MS Analysis

According to SANTE guidelines, confirmation of samples should be performed by MS detector. GC/MS analysis was performed with similar conditions of GC-ECD only in the positive samples observed in GC-ECD in order to have confirmation. GC/MS instrument, TRACE GC Ultra (Thermo Fisher Scientific, Austin, TX, USA) gas chromatograph coupled with a Polaris Q ion trap mass spectrometer was used. The transfer line and the ion source temperature were 260 and 270 °C, respectively. Data acquisition was performed first in full scanning mode from 50 to 500 *m*/*z* to confirm the retention times of the analytes. All standards and sample extracts were analyzed in selective ion monitoring (SIM) mode. PCB101 confirmation was performed with the identification of three *m*/*z* ions 326 > 324 > 286. 

### 2.5. Statistical Analysis

Two-way ANOVA statistical analysis was applied to estimate significant differences among different analytical procedures using GraphPad software. Multiple comparisons were performed where each mean value was compared to each group of contaminants. 

## 3. Results and Discussion

The extraction and clean-up steps for kiwano’ matrices were a challenging part of the method development due to its rich composition in carotenoids, steroids, alkaloids, saponins, glycosides, flavonoids, tannins, and phenolic compounds [1,3]. The optimization of analytical methods for the determination of 30 contaminants in kiwano samples included the two crucial steps of the QuEChERS procedure: (1) Sample extraction and (2) the d-SPE clean-up. Figure 2 shows the chromatogram obtained when the mixture of the 30 multiple contaminants was analyzed by GC-ECD and FPD in the method described previously in Section 2.4.1.1 and Section 2.4.2. The extraction recovery of the method was evaluated by spiking the kiwano sample with the multiple contaminant solutions at 7.5 µg kg^−1^. Four protocols were tested: (1) QuEChERS AOAC with additional d-SPE clean-up CL1 (150 mg of MgSO_4_, 50 mg of PSA, and 50 mg of GCB), (2) QuEChERS AOAC with additional d-SPE clean-up CL2 (150 mg of MgSO_4_, 50 mg of PSA, and 25 mg of GCB), (3) QuEChERS EN with additional d-SPE clean-up CL1, and (4) QuEChERS EN with additional d-SPE clean-up CL2. 

The study of the evaluation of the method’s efficiency was carried out according to the guidelines of the SANTE document [38], being the range of recovery established 70 to 120%. In Figure 3, poor extraction recoveries were observed for some of the chemical families using QuEChERS AOAC. The OCP, PCB, and BFR compounds presented recoveries of less than 70% using the QuEChERS AOAC and CL1, while for QuEChERS AOAC and CL2 only the PCB compounds. Since recovery percentages after the clean-up CL1 (150 mg of MgSO_4_, 50 mg of PSA, and 50 mg of GCB) for QuEChERS AOAC evaluation were not satisfactory, the approach testing test other QuEChERS contents (EN) and another d-SPE clean-up (CL2) was followed. After reducing GCB in the CL2 clean-up and using QuEChERS EN, an improvement in extraction recoveries for all targeted multiple compounds was stated. The most evident result on extraction efficiency is the negative influence of the amount of GCB used in the second step of the extraction. As previously reported, GCB adsorbs compounds such as pigments, anthocyanins, and carotenoids, as well as planar compounds [23,33]. Therefore, reducing its quantity in the cleaning step is one of the optimizations of this process. Although the lower amount of GCB did not absorb all the coloring compounds like the previous CL1 clean-up, the samples were still suitable for GC analysis. ANOVA statistical analysis was used to compare the mean recoveries of each cleaning test (CL1, CL2) between the target chemical groups (OCP, OPP, PCB, BFR). The two-way ANOVA statistical study showed that the recoveries are significantly different comparing the two different clean-up sets (CL1 and CL2) for OCP and BFR using QuEChERS AOAC while for QuEChERS EN all chemical groups were statistically different. Overall, the results showed that most of the compounds are in the 70–120% range when QuEChERS EN and CL2 are used. Figure 4 shows a summary of the results of the recovery studies. It was observed that in the satisfactory range 70–120%, the highest number of contaminants was achieved with QuEChERS EN and CL2.

As previously reported, a detailed optimization is an extremely important step as it reveals which compounds show the best results. As reported by Fernandes et al. [22,23,24,35], this extraction method is suitable but needs to be optimized and studied for each group of compounds and matrices.

The results, displayed in Figure 3 and Figure 4, allowed us to assess that the best extraction and cleaning procedures for kiwano were QuEChERS EN with a clean-up CL2 (150 mg of MgSO_4_, 50 mg of PSA, and 25 mg of GCB), and this was selected for all further investigations.

### 3.1. Matrix Effects

In the present work, the matrix effect was evaluated by comparing the slope obtained with the calibration curves of each compound in the matrix phase and *n*-hexane. This evaluation was complemented by comparing the retention times of the chromatograms with the same concentration in the matrix phase and *n*-hexane, and no significant differences were observed. It is well described in the literature that some analytes in fruit extracts exhibit a matrix signal enhancement/suppression effect when analyzed by GC [23,39]. This effect occurs when interferences from fruit matrices (such as pigments, lipids, acids, etc.) compete with the target analytes in the GC injector [40]. Figure 5 shows that the different chemical families (OCP, OPP, PCB, and BFR) analyzed in kiwano fruits presented different matrix effects behaviors. The signal enhancement was observed with the use of both QuEChERS AOAC and EN with the CL2 cleaning step. Additionally, with QuEChERS AOAC and CL2 clean-up, the mean matrix factor value was higher than 1.2 in all the chemical families. The BFR are those with the highest signal increase. The QuEChERS EN showed a satisfactory matrix factor with CL1 clean-up. However, as shown in Section 3, the extraction efficiency was not acceptable with this extraction procedure. In any case, this study confirmed that the matrix effect was more evident when the lowest amount of GCB sorbent was used. 

### 3.2. Method Validation

Method validation is an important requirement in the practice of an analytical method process. The reliability and robustness of the method to be used for real sample analysis should be studied considering several analytical parameters. Linearity, extraction recovery at three spiking levels (7.5, 11.2, 14.9 µg kg^−1^), precision, LODs and LOQs obtained by the regression analysis (based on the standard deviation of the response of the curve and the slope of the calibration curve), as well as matrix effects, were the parameters studied for the validation of analysis of multiple contaminants in kiwano samples. Table 1 summarizes the analytical parameters in order of retention time obtained by GC-ECD and GC-FPD. 

Considering the matrix effects described in the previous section, the analytical validation process was carried out in kiwano extract. Matrix-matched calibration curves were obtained in kiwano extracts of the 30 target analytes with a coefficient of determinations greater than 0.991. LODs and LOQs ranged from 0.2 to 2.2 and 0.6 to 7.4 µg kg^−1^, respectively (Table 1). The mean recoveries at the three spiking levels of 7.5, 11.2, and 14.9 µg kg^−1^ ranged from 90% and 122% (99% on average) with relative standard deviation (RSD) values between 8% and 15%. The method precision was determined through intra-day and inter-day repeatability experiments by five repeated measurements, and the results were less than 15% of RSD, which is suggested as the acceptable precision (Table 1). When compared to other studies on exotic fruits [30], we can say that for organochlorine pesticides, for example, the analytical parameters, namely the LOD and LOQ, are much better in the present work. As for the BFR, a study in *capsicum* cultivars [23] already reported presents higher LOD and LOQ values than those obtained for Kiwano.

Although the European Union legislation for pesticides [41] does not include the kiwano fruit, the analytical parameters obtained for this method meet the requirements. As for the other studied compounds, most of them are not included in the food legislation, despite being frequently detected in food products. As an example, EFSA recommends BFR monitoring studies in food samples [42].

### 3.3. Kiwano Sample Analysis

After the method validation, the optimized method was applied to evaluate possible contamination in kiwano samples. Since the study was carried out on the kiwano pulp, as it is the edible part, the results are presented by pulp mass. The screening of the 30 multiple contaminants in a total of 10 kiwano samples led to the identification and quantification of PCB 101 (5.1 µg kg^−1^ in the kiwano pulp) in a single sample. GC/MS analysis confirmed the presence of PCB 101 (Figure 6). It was also confirmed that, except for one sample, the kiwano fruit samples are safe in terms of 12 OCP, 6 OPP, 7 BFR, and 5 PCB studied. The presence of pesticides is well reported in the literature on fruits [28,43,44,45], concerning other contaminants, the works are less represented. However, PCBs, mostly associated with anthropogenic sources, have been reported in grapes, and other several fruits [46,47] and BFR in red fruits [24], *capsicum* cultivars [23], among others [48]. This work was performed in a small number of samples, and Portugal is still in the beginning regarding this crop. However, it shows the great importance of including these fruits in monitoring studies and that it should be extended to a larger number of samples from different production sites. Furthermore, the results suggest the importance of including other organic contaminants in monitoring studies on food samples in addition to pesticides.

## 4. Conclusions

An analytical methodology based on an optimized QuEChERS technique was effectively applied for the simultaneous analysis of 30 multiple contaminants (12 OCP, 7 OPP, 5 PCB, and 7 BFR) in kiwano samples. The optimized QuEChERS procedure encompassed the study of two QuEChERS compositions (QuEChERS AOAC and EN) in addition to two d-SPE clean-up compositions (CL1 and CL2). Although matrix effects were observed, it was found that QuEChERS EN, in combination with CL2 clean-up, offered an improvement in overall extraction recovery of the multiple target contaminants. Based on these results, it can be concluded that analytical method optimization studies are crucial for the analysis of multiple compounds in complex matrices. The methodology meets the analytical requirements in terms of accuracy, sensitivity, and precision. The novelty of this study allows the evaluation of multiple contaminants in kiwano samples, ensuring their safe commercialization in terms of the presence of pesticides and other organic contaminants. The presence of PCB 101 in one kiwano fruit reinforces the need for monitoring studies of organic contaminants, such as PCBs and BFRs.

## Figures and Tables

**Figure 1 foods-12-00993-f001:**
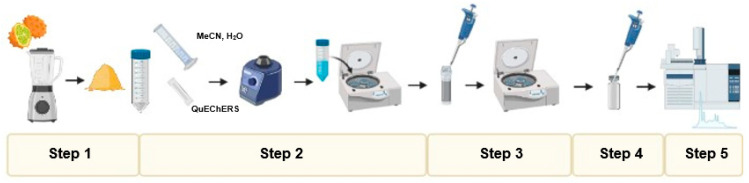
Scheme of the experimental procedure.

**Figure 2 foods-12-00993-f002:**
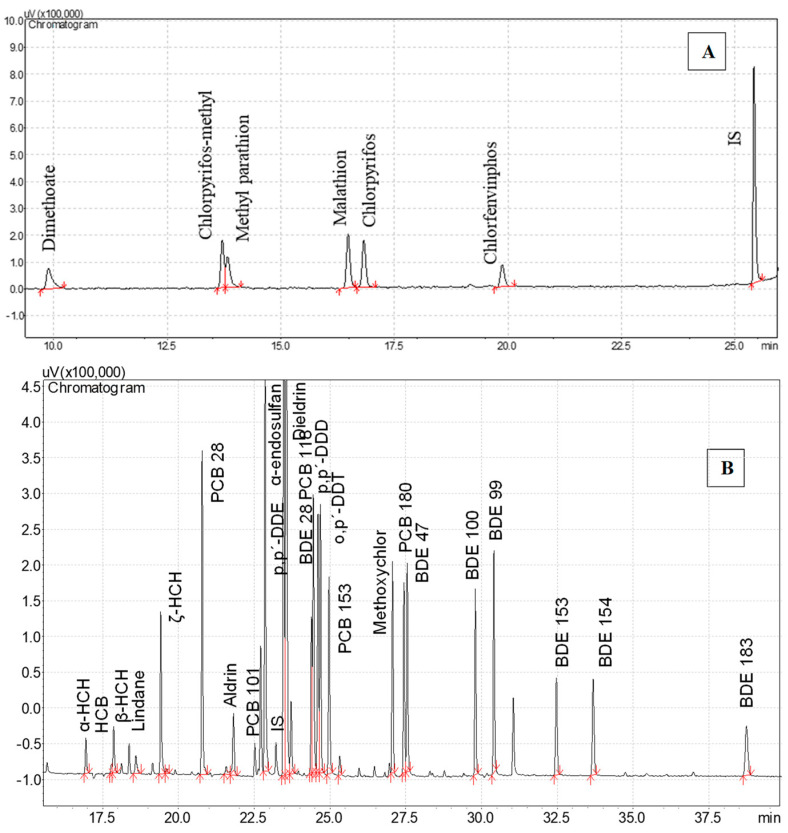
(**A**) Chromatogram of the injection by GC-FPD of a standard mixture of 7 organophosphorus pesticides at 7.5 µg kg^−1^. (**B**) Chromatogram of the injection by GC-ECD of a standard mixture of 23 halogenated organic compounds (5 PCB, 7 BFR and 12 OCP) 7.5 µg kg^−1^.

**Figure 3 foods-12-00993-f003:**
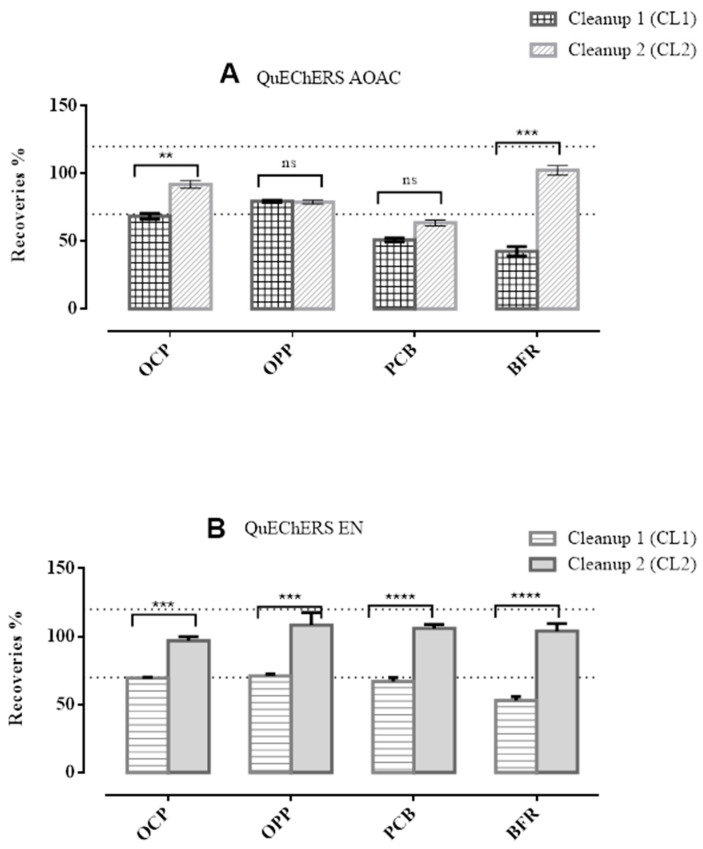
Mean extraction recoveries (%) of the targeted multiple contaminants divided into four chemicals groups (OCP, *p*-value = 0.006; OPP, *p*-value = 1; PCB, *p*-value = 0.06 and BFR, *p*-value = 0.0002) using QuEChERS AOAC (**A**) and QuEChERS EN (**B**). Two-way ANOVA analysis with Sidak’s multiple comparisons test (ns: Non-significant; **/***/****—significant).

**Figure 4 foods-12-00993-f004:**
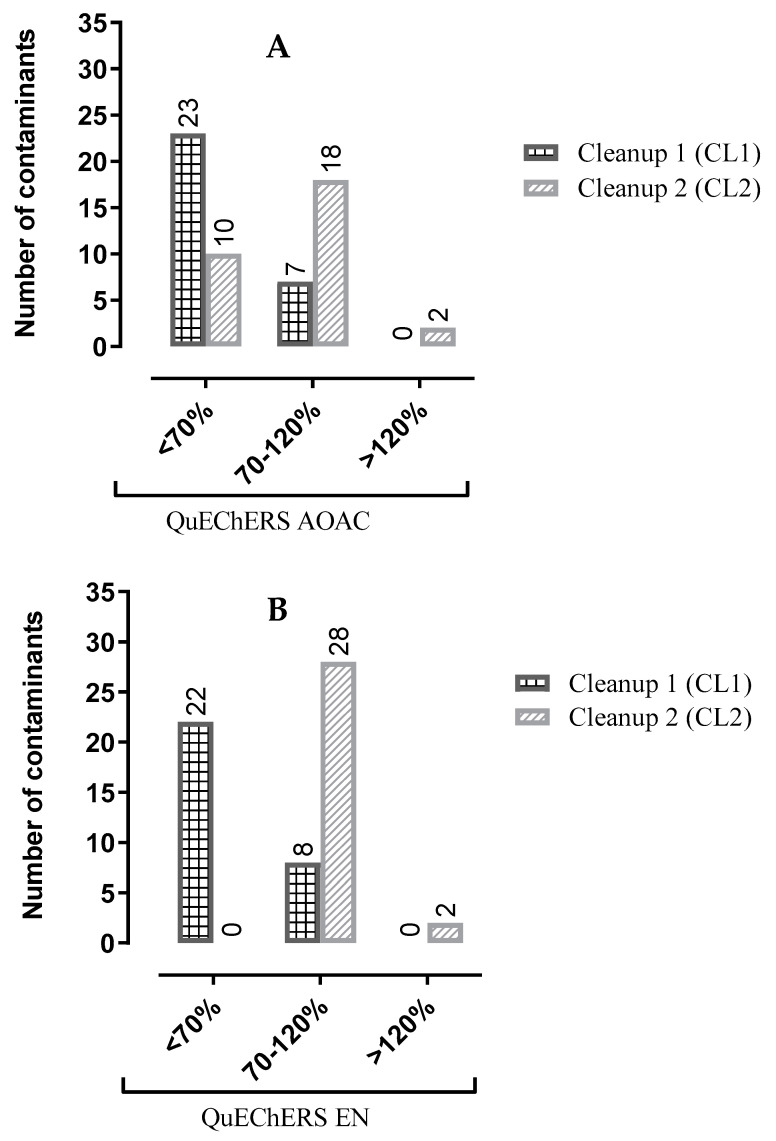
Number of contaminants obtained in the ranges lower than 70%, between 70 and 120% and higher than 120%, using QuEChERS AOAC (**A**) and EN (**B**).

**Figure 5 foods-12-00993-f005:**
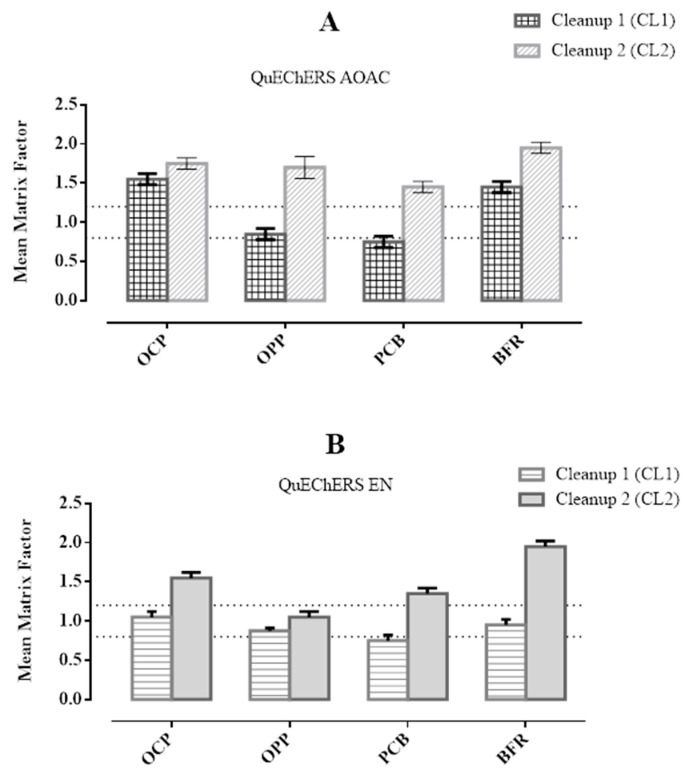
Matrix factor results for the chemical families studied using QuEChERS AOAC (**A**) and EN (**B**).

**Figure 6 foods-12-00993-f006:**
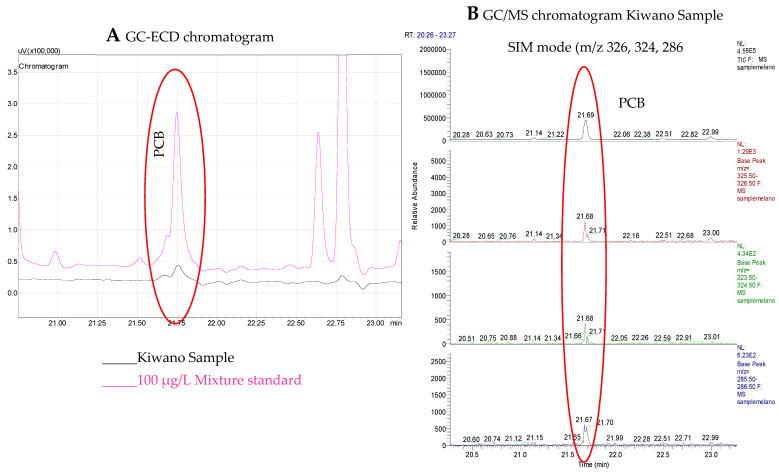
(**A**) GC-ECD chromatogram overlapping kiwano sample and 100 mg L^−1^ mixture standard solution and (**B**) GC-MS spectrum of confirmation of the presence of PCB 101 (*m*/*z* 286, 324, 326).

**Table 1 foods-12-00993-t001:** Data including correlation to the matrix-matched calibration curve, the limit of detection (LOD) and limit of quantification (LOQ), mean recoveries (from three spiking levels), and precision obtained for the 30 target contaminants.

	Linearity Range µg kg^−1^	Coefficient of Determination	LOD µg kg^−1^	LOQ µg kg^−1^	Mean Recovery (*n* = 3)%	Precision (n = 5) Intra-Day Inter-Day %	
α-HCH	2.2–18.7	0.9922	2.1	6.9	95	10	14
HCB	2.2–18.7	0.9933	2.0	6.6	94	14	15
β-HCH	2.2–14.9	0.9936	2.2	7.3	90	9	11
Lindane	2.2–14.9	0.9941	2.1	7.1	103	9	13
ζ-HCH	2.2–14.9	0.9926	1.8	5.9	91	8	11
PCB 28	2.2–18.7	0.9928	2.2	7.3	90	9	10
Aldrin	1.5–18.7	0.9989	1.2	4.2	99	8	9
PCB 101	1.5–18.7	0.9986	1.4	4.7	110	10	12
End I	1.5–18.7	0.9995	0.8	2.8	105	12	15
p,p′-DDE	2.2–18.7	0.9939	1.7	5.6	99	8	9
Dieldrin	1.5–18.7	0.9990	1.1	3.6	114	8	10
PCB 118	1.5–18.7	0.9987	1.1	3.7	90	9	12
BDE 28	1.5–18.7	0.9999	0.3	1.0	99	8	11
p,p′-DDD	1.5–18.7	0.9998	0.2	0.6	91	10	14
o,p′-DDT	1.5–18.7	0.9992	1.0	3.2	90	13	15
PCB 153	1.5–18.7	0.9996	0.7	2.2	106	9	11
Methoxychlor	2.2–18.7	0.9965	1.9	6.3	93	9	14
PCB 180	1.5–18.7	0.9983	1.3	4.3	122	8	9
BDE 47	1.5–18.7	0.9995	0.7	2.3	108	8	10
BDE 100	2.2–18.7	0.9929	2.0	6.6	99	13	15
BDE 99	1.5–18.7	0.9990	1.0	3.3	103	9	12
BDE 153	1.5–18.7	0.9993	0.8	2.8	122	7	9
BDE 154	1.5–18.7	0.9998	0.4	1.4	95	6	9
BDE 183	1.5–18.7	0.9995	0.7	2.3	91	9	12
Dimethoate	2.2–18.7	0.9938	1.9	6.3	90	8	11
Chlorpyrifos-methyl	2.2–18.7	0.9931	2.2	7.3	93	7	9
Methyl parathion	2.2–18.7	0.9960	1.7	5.7	105	9	10
Malathion	2.2–18.7	0.9929	2.0	6.6	94	10	13
Chlorpyrifos	1.5–18.7	0.9993	0.9	2.9	97	8	10
Chlorfenvinphos	1.5–18.7	0.9989	1.1	3.6	90	13	15

## Data Availability

The data are available from the corresponding author.

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
