# Peer review of "Multiple Organic Contaminants Determination Including Multiclass of Pesticides, Polychlorinated Biphenyls, and Brominated Flame Retardants in Portuguese Kiwano Fruits by Gas Chromatography"

_foods, 2023, doi:10.3390/foods12050993_

Round 1

Reviewer 1 Report

Although the article is dealing with a subject of relatively increased scientific merit, the article seems to have been submitted without the required attention to writing details. It reminds more of a draft submitted under pressure of time.

Author Response

Response to Reviewer 1 Comments

Point 1: Although the article is dealing with a subject of relatively increased scientific merit, the article seems to have been submitted without the required attention to writing details. It reminds more of a draft submitted under pressure of time.

 Response 1: Thank you very much for your comment. The article has been improved with the various suggestions from the reviewers and we hope that the new version is more thoughtful and improved. The english and grammar, as well as other relevant aspects have been improved.

Reviewer 2 Report

The title of the manuscript is “Multiple Organic Contaminants Determination Including Multiclass of Pesticides, Polychlorinated Biphenyls, and Brominated Flame Retardants in Portuguese Kiwano fruits By Gas Chromatography.” Kiwano is a tropical fruit, and this is the first case where OCP, OPP, PCB, BFR are analyzed. Statistical techniques were used to logically compare various preprocessing methods. Some minor grammatical errors or typos may need to be corrected in order for this article to be considered suitable for publication in this journal. Some revisions are described below.

 1.          Please modify the abstract to be within 200 characters and its format in accordance with the guidelines of the journal. The guidelines for the abstract of Foods are as follows:

“Abstract: The abstract should be a total of about 200 words maximum. The abstract should be a single paragraph and should follow the style of structured abstracts, but without headings.”

2.          Line 42-87: The paragraph is too long. Please divide it into appropriate topics for distinction.

3.          Line 193: Please change "The GC analysis were" to "The GC analysis was".

4.          Line 199: Change “(m/z)” to “(mass to charge ratio; m/z)”

5.          Line 260: Please describe in detail what "the method described previously" in the sentence "GC-ECD and FPD in the method described previously." refers to.

6.          Figure 2: Traditional GC should separate all of the analytes perfectly. However, as seen in the figure 2(a), some components are not separated at all (e.g. chlorpyrifos-methyl and methyl parathion). In this case, proper quantitation is not possible. Also, in the Figure 2(b), it is not confirmed whether some analytes have been accurately separated. Authors need to add or modify the chromatogram by enlarging it to confirm the separation.

7.          Please add labels (numbers) on top of the bars in Figure 4.

8.          Line 318-340: This paragraph is not about validation of analytical method; it is about the comparison results of preprocessing methods for matrix effect. I suggest adding a new section specifically for this topic.

9.          Figure 5: Matrix effect can also be affected by retention time (tR). Have you investigated the matrix effect according to the distribution of tR?

10.      Line 372-387: Is PCB 101 the only analyte detected in one sample? Please clearly specify how many other real samples were analyzed and whether any other analytes were detected in these samples.

Author Response

Response to Reviewer 2 Comments

General comments:

The title of the manuscript is “Multiple Organic Contaminants Determination Including Multiclass of Pesticides, Polychlorinated Biphenyls, and Brominated Flame Retardants in Portuguese Kiwano fruits By Gas Chromatography.” Kiwano is a tropical fruit, and this is the first case where OCP, OPP, PCB, BFR are analyzed. Statistical techniques were used to logically compare various preprocessing methods. Some minor grammatical errors or typos may need to be corrected in order for this article to be considered suitable for publication in this journal. Some revisions are described below.

Response : Thank you very much for the comments that have received our best attention and have greatly improved the paper.

Point 1: Please modify the abstract to be within 200 characters and its format in accordance with the guidelines of the journal. The guidelines for the abstract of Foods are as follows:

“Abstract: The abstract should be a total of about 200 words maximum. The abstract should be a single paragraph and should follow the style of structured abstracts, but without headings.”

Response 1: The abstract was revised and and with a total of 200 words. 

Point 2: Line 42-87: The paragraph is too long. Please divide it into appropriate topics for distinction.

Response 2: Thank you very much for the suggestion. The paragraph was divided in different topics. 

Point 3: Line 193: Please change "The GC analysis were" to "The GC analysis was".

Response 3: Thank you very much for the correction. The text has been changed.

Point 4: Line 199: Change “(m/z)” to “(mass to charge ratio; m/z)”

Response 4: Thank you very much for the suggestion. The text has been changed.

Point 5: Line 260: Please describe in detail what "the method described previously" in the sentence "GC-ECD and FPD in the method described previously." refers to.

Response 5: Thank you very much for the suggestion. The sentence was clarified in the manuscript (line 240)

Point 6: Figure 2: Traditional GC should separate all of the analytes perfectly. However, as seen in the figure 2(a), some components are not separated at all (e.g. chlorpyrifos-methyl and methyl parathion). In this case, proper quantitation is not possible. Also, in the Figure 2(b), it is not confirmed whether some analytes have been accurately separated. Authors need to add or modify the chromatogram by enlarging it to confirm the separation.

Response 6: Thank you very much for your comment. It is true that we have some compounds where the separation is not perfect. It is not possible to separate them. Since chromatography is a separation technique, we always make the methods in order to analyze as many compounds as possible at the same time. And in this case, out of 30 compounds only 6 are not completely separated.

However, it is quite visible their separation and the splitting of the peaks is done rigorously in order to allow their quantification. The chromatographic analyses are performed in triplicate, and the precision between analyses is always evaluated. The authors are of the opinion that these compounds should not be excluded from the study for this reason.  

Point 7: Please add labels (numbers) on top of the bars in Figure 4.

Response 7: Thank you very much for the suggestion. The numbers on the top of the bars in Fig 4 were added. The authors also detected an error in the figure and so, the graphics were also corrected. .

Point 8: Line 318-340: This paragraph is not about validation of analytical method; it is about the comparison results of preprocessing methods for matrix effect. I suggest adding a new section specifically for this topic.

Response 8: Thank you very much for the suggestion. The matrix effects study was separated in a new section (3.2).

Point 9: Figure 5: Matrix effect can also be affected by retention time (tR). Have you investigated the matrix effect according to the distribution of tR?

Response 9: Thank you very much for the comment. We compared overlap chromatograms and no significant difference in retention times has been verified. This result was added to the manuscript (Line 299-302)

Point 10: Line 372-387: Is PCB 101 the only analyte detected in one sample? Please clearly specify how many other real samples were analyzed and whether any other analytes were detected in these samples.

Response 10: Thank you very much for the comment. It was clarified that the evaluation of contaminants was performed on 10 samples.

Reviewer 3 Report

Comment: This is an interesting study and the authors have developed an analytical method based on the QuEChERS technique for the evaluation of 30 multiple contaminants in Portuguese Kiwano fruits. Although the data in the text supports the conclusions and this paper has a potential to be accepted, but some points have to be clarified or fixed before we can proceed and a positive action can be taken.

1-      The manuscript needs revision for grammar. Some parts of the manuscript need to editing. For example, in section 3. line 262 needs to editing (what is four extraction clean up?). It may be “for extraction and clean up, four ……”

2-      I suggest major rewrite of the line 274-281 and line 384.

3-      Section 2.6, line 238 to 252 are not related to the paper.

Author Response

Response to Reviewer 3 Comments

General Comment: This is an interesting study and the authors have developed an analytical method based on the QuEChERS technique for the evaluation of 30 multiple contaminants in Portuguese Kiwano fruits. Although the data in the text supports the conclusions and this paper has a potential to be accepted, but some points have to be clarified or fixed before we can proceed and a positive action can be taken.

Response : Thank you very much for your comment. The article has been improved with the various suggestions from the reviewers and we believe that the revised version is improved and suitable for publication.

Point 1: The manuscript needs revision for grammar. Some parts of the manuscript need to editing. For example, in section 3. line 262 needs to editing (what is four extraction clean up?). It may be “for extraction and clean up, four ……”

Response 1: Thank you very much for your comment. The manuscript was revised and improved in English and grammar as well as other relevant aspects.

Point 2: I suggest major rewrite of the line 274-281 and line 384.

Response 2: Thank you very much for your comment. The paragraph and the sentence in line 384 were rewritten.

Point 3: Section 2.6, line 238 to 252 are not related to the paper.

Response 3: The authors apologize. The lines in question are from the Foods word template that by mistake were not deleted. This text has been deleted.
